# Unscented Kalman Filter-Based Robust State and Parameter Estimation for Free Radical Polymerization of Styrene with Variable Parameters

**DOI:** 10.3390/polym14050973

**Published:** 2022-02-28

**Authors:** Zhenhui Zhang, Zhengjiang Zhang, Zhihui Hong

**Affiliations:** National-Local Joint Engineering Laboratory for Digitalize Electrical Design Technology, College of Electrical and Electronic Engineering, Wenzhou University, Wenzhou 325035, China; zhzhang519@163.com (Z.Z.); zzjiang108@gmail.com (Z.Z.)

**Keywords:** state and parameter estimation, unscented Kalman filter, RMSProp gradient descent, polymerization

## Abstract

The free radical polymerization of styrene (FRPS) is a complex process system with uncertain parameters in its mechanistic model. When the reaction conditions are switched, or the reaction process generates faults, the parameters will change. Therefore, state and parameter estimation (SPE) becomes an important part of the process monitoring and process control for free radical polymerization of styrene. The unscented Kalman filter (UKF) is widely used for nonlinear process systems, but it rarely considers the problem of model parameter uncertainty. UKF can be used for SPE, called UKF-based SPE (UKF-SPE), where the parameters are usually estimated simultaneously as an extension of the state space. However, when the parameters change with system switching, the traditional UKF-SPE cannot detect and track the parameter changes in time, and inaccurate parameters generate modeling errors. To deal with the problem, a UKF-based robust SPE method (UKF-RSPE) for the free radical polymerization of styrene with variable parameters is proposed, introducing a parameter testing criterion based on hypothesis testing and moving windows to directly detect whether the parameters have changed. Based on the detection results, a gradient descent method with adaptive learning rate is used to iteratively update the parameters to speed up the tracking of the parameters and to obtain more accurate parameters and states. Finally, the proposed UKF-based robust SPE is applied to free radical polymerization of styrene in a jacketed continuous stirred tank reactor. The experimental results verify the effectiveness and robustness of the method, which can track the parameters faster and obtain more accurate states.

## 1. Introduction

The output of the free radical polymerization of styrene is distributed, called distribution output, and the control of the distribution output is critical because it significantly affects product quality. Therefore, accurate estimation of states and parameters is important to monitor the process of free radical polymerization of styrene. Models of free radical polymerization of styrene usually generate modeling errors due to uncertain parameters. In order to obtain accurate states and parameters, state and parameter estimation (SPE) plays an important role in the free radical polymerization of styrene.

State estimation, also called filtering, uses the redundancy of a real-time measurement system to improve data accuracy, automatically eliminates error messages caused by random disturbances, and can estimate states. Kalman proposed the Kalman filter (KF) in 1960, which was restricted to linear systems [1,2]. Subsequently, Bucy et al. proposed the extended Kalman filter (EKF) by linearizing the nonlinear equations to first order through Taylor expansions, but it inevitably introduced approximation errors [3]. In 1999, Julier et al. proposed the unscented Kalman filter (UKF) based on the unscented transformation (UT) to overcome the problem of EKF, which is widely used for nonlinear systems [4].

The free radical polymerization process of styrene is a large nonlinear dynamic model. Already in 1981, Schmidt and Ray proposed a mathematical model for the dynamics of polymerization in a jacketed continuous stirred tank reactor (CSTR). The model contains the material balance equation for different molecular weight polymer concentrations and derives the differential equation for the heavy mean molecular weight [5]. The state estimation of the free radical polymerization of styrene allows to accurately grasp the reaction process, predict the trend of the reaction, and reconcile the measured data. Based on this model, Zhang and Chen developed two data-processing algorithms, including correntropy-based nonlinear dynamic data reconciliation (CNDDR) and gross error detection and identification (GEDI), which greatly reduced the effect of gross errors on the reconciliation results of free radical polymerization of styrene [6]. Hu et al. proposed a robust EKF combined with measurement compensation (MC-REKF), and applied it to this model to compensate for gross errors in measurements and obtained accurate states [7].

Since the process model of free radical polymerization of styrene is complex, many modeling approaches are now available. Hu et al. effectively combined Elman neural network (ENN) with EKF to form a data-driven dynamic data coordination scheme, called ENN-EKF, which is able to model the free radical polymerization of styrene and effectively suppress the effect of measurement noise [8]. Al-Harthi et al. developed a comprehensive mathematical model for atom transfer radical copolymerization reactions in intermittent reactors [9]. Mohammadi et al. modeled styrene radical polymerization in a stirred tank reactor-spinning disc reactor (STR-SDR) device using Kinetic Monte Carlo (KMC) algorithm [10]. Maafa et al. developed a dynamic Monte Carlo model capable of simulating the polymerization of styrene with a bifunctional radical initiator in an intermittent reactor [11]. Woloszyn et al. developed a styrene polymerization model with bifunctional free-radical initiators in a batch reactor and considered the effect of parameters on model predictions [12]. Meißner et al. used machine learning to identify parameters [13]. Uncertainty of parameters, a problem in modeling, will have a huge impact on the mechanistic modeling of the free radical polymerization of styrene.

In order to obtain the accurate parameters of the mechanism model, a number of KF-based parameter estimation methods have been developed and are used in academia and industry. Kiparissides et al. used EKF to estimate the state variables and time-varying termination rate constants of the model [14]. Yousefi-Darani et al. used UKF for online estimation of biomass, glucose, and ethanol concentrations, and growth rate parameters [15]. Zhou et al. proposed a state-input-parameter mixed KF (SIPMKF) to deal with the coexistence of uncertain model parameters and control inputs [16]. Kim et al. proposed a method in estimating important parameters of gray-box model based on real nonlinear electroencephalogram (EEG) [17]. Geraldi et al. proposed a new UKF method for estimating synchronous generator parameters using measurements under unbalanced operating conditions [18]. Peng et al. proposed a battery modeling and online parameter identification method based on the Davinan equivalent circuit model (ECM) and recursive least squares (RLS) algorithm with forgetting factor [19]. Kong et al. proposed a new parameter adaptive method aging for state of charge (SOC) estimation [20]. Nakamura and Kawamura used the UKF to estimate the heat capacity and solar radiation shadow coefficients [21]. Li et al. used a joint state parameter UKF estimation method by adding state vectors to include model parameters to estimate the unknown states and parameters [22]. Chen et al. used UKF to estimate model parameters of a resistance-capacitance (RC) model [23]. Rodríguez et al. proposed a dual KF, which is used for state estimation and parameter estimation [24]. Wan et al. proposed a new UKF algorithm combined with the Huber method [25]. Lee and Song proposed a dual adaptive UKF to identify parameters of a dynamic system [26]. Ding et al. proposed a new Bayesian adaptive UKF algorithm for the aerodynamic parameter and noise estimation for aircraft dynamical systems [27]. Although the above scholars have studied the application and improvement of KF in parameter estimation, they have not been used in free radical polymerization of styrene with variable parameters. Most of the above studies used the method of state and parameter estimation, which is seen as an extension of the state by setting the parameters as time-varying parameters, and this method is called artificial evolution, and this strategy is widely used in SPE techniques [28,29].

The traditional SPE with slow tracking parameters will lead to slow convergence of the measurement error, making accurate parameter estimation difficult. Wang et al. proposed a new adaptive law to achieve optimal parameter estimation to maintain simultaneous convergence of the estimation error and tracking error [30]. To accomplish accurate and fast parameter estimation, gradient descent methods with adaptive learning rate are used to optimize filters. Common methods of adaptive learning rate include Delta-bar-delta [31], AdaGrad [32], RMSProp (Root-Mean-Square-Prop), Adam [33], etc. Long et al. proposed an adaptive online parameter identification method based on gradient descent optimization (GDO) and a variable iteration step search method based on the RMSProp gradient descent optimization (RMSprop-GDO) method to reduce the search time required for the optimal iteration step [34]. Long et al. proposed an online gradient learning algorithm with adaptive learning rate for identifying the parameters of a neuro-fuzzy system with a Gaussian fuzzy set representing the Mamdani fuzzy model [35]. Na et al. proposed an adaptive parameter estimation framework for the nonlinear systems with time-varying parameters [36].

Although the influence of parameters in the mechanistic model of free radical polymerization of styrene is taken into account, the state of the reaction process will change with the variation of parameters when the operating conditions of the reaction are changed, or some failure occurs in the reaction. Therefore, parameters and states need to be estimated together to obtain an accurate system model and estimated states. Because of the limitations of artificial evolution, these parameters cannot be tracked quickly and accurately by traditional SPE. In this paper, a UKF-based robust SPE method for free radical polymerization of styrene with variable parameters is proposed. A hypothesis test-based parameter testing criterion is introduced to directly identify whether the parameters change. Based on the identification results, the learning rate of the gradient descent method is adaptively modified, and the gradient descent method is used to accelerate the parameter tracking. Finally, accurate parameters and states are obtained simultaneously.

The remainder of the paper is organized as follows. The foundations of UKF-based SPE are reviewed in Section 2. Section 2.1 introduces the algorithm of standard UKF for state estimation, and Section 2.1 introduces UKF-based SPE. In Section 3, the UKF-based robust SPE scheme is proposed to improve parameter estimation performance, which combines moving window and adaptive learning rate gradient descent. The proposed UKF-based robust SPE is applied to free radical polymerization of styrene with variable parameters, and the results demonstrate the effectiveness the proposed UKF-based robust SPE in Section 4. Finally, the conclusions are drawn in Section 5.

## 2. Foundations of UKF-Based SPE

### 2.1. Review of Standard UKF

KF is a model-based linear minimum variance estimation, which is widely used in stochastic signal processing, but it is only applicable to linear systems. EKF solves this problem, but it must do a Taylor expansion of the nonlinear system equations and retain only the linear term, so EKF is only applicable to weakly nonlinear systems, and the stronger the nonlinearity, the larger the estimation error. To solve the estimation problem of nonlinear systems, UKF is proposed. UKF is based on the minimum variance estimation of the model like KF, while the model of UKF is a nonlinear system. The standard state space model of nonlinear dynamic system containing state equation and measurement equation has a discrete form, which can be described as
(1)$$x_{k}=f(x_{k-1},u_{k-1},\theta )+w_{k-1}$$
(2)$$z_{k}=h(x_{k},u_{k},\theta )+v_{k}$$
where k denotes the time step of the system operation. $x_{k}\in R^{N_{x}\times 1}$ is the process state of system, which is usually unknown during system operation; $z_{k}\in R^{N_{z}\times 1}$ is the vector of measurement; and $u_{k-1}$ is the vector of input. $f=[f_{1};f_{2};\cdots ;f_{N_{x}}]$ and $h=[h_{1};h_{2};\cdots ;h_{N_{z}}]$ are the state equations and measurement equations, respectively. $w_{k-1}$ and $v_{k}$ are the vectors of uncorrelated zero-mean white noise, i.e., $E\left\{w_{i}v_{j}^{T}\right\}=0,\forall i,j$. The process noise $w_{k-1}$ and the measurement noise $v_{k}$ are assumed to Gaussian distribution, i.e., $w_{k-1}\sim G(w_{k-1};0,Q)$, and $v_{k}\sim G(v_{k};0,R)$, the process noise covariance matrix Q and measurement noise covariance matrix R depend on the sensor accuracy and the interference of environmental factors in the system, respectively. Q and R are then set as diagonal matrices and are assumed to be constant in the present work. θ is the vector of constants parameters. For traditional state estimation, the parameters are known constants, but as the system becomes more complex, uncertain parameters may appear in the model. This is also the focus of this paper’s research.

#### 2.1.1. Unscented Transformation

The major difference between UKF and KF is the finding of the optimal gain array, which depends on the mean and covariance. The UKF uses the unscented transformation (UT) to deal with the nonlinear transfer problem of the mean and covariance [4]. In Figure 1, the UT generates Sigma points in the original state distribution based on certain rules, and the mean and covariance of these points are equal to the original state distribution, and then the nonlinear system equations are used for nonlinear transfer to obtain the set of nonlinear function values, which leads to the transformed mean and covariance. Suppose the random vector X is nonlinearly transformed by f(·) to obtain the random vector Y, i.e., Y=f(X). If the mean $\bar{X}$ and covariance matrix $P_{XX}$ of X is known, the mean $\bar{Y}$ and covariance matrix $P_{YY}$ of Y can be calculated by UT. The detailed steps are as follows.
(3)$$\chi^{\left(i\right)}=\left\{ \begin{array}{l} \bar{X},i=0\\ \bar{X}+P_{XX,i},i=1\sim 2N_{x} \end{array}\right.\;\sqrt{(N_{x}+\lambda )P_{XX}}=[p_{1},p_{2},\cdots ,p_{N_{x}}]\;-\sqrt{(N_{x}+\lambda )P_{XX}}=[p_{N_{x}+1},p_{N_{x}+2},\cdots ,p_{2N_{x}}]$$

The sigma points are transferred through f(·) to obtain the transformed sigma points $Y^{\left(i\right)}$.
(4)$$Y^{\left(i\right)}=f(\chi^{\left(i\right)}),i=0\sim 2N_{x}$$

Then, the weights are determined according to the symmetric sampling rule as
(5)$$W_{m}^{\left(i\right)}=\left\{ \begin{array}{l} \frac{\lambda }{N_{x}+\lambda },i=0\\ \frac{1}{2(N_{x}+\lambda )},i=1\sim 2N_{x} \end{array}\right.$$
(6)$$W_{c}^{\left(i\right)}=\left\{ \begin{array}{l} \frac{\lambda }{N_{x}+\lambda }+(1-\alpha^{2}+\beta ),i=0\\ \frac{1}{2(N_{x}+\lambda )},i=1\sim 2N_{x} \end{array}\right.$$
(7)$$\lambda =\alpha^{2}(N_{x}+\kappa )-N_{x}$$
where $N_{x}$ donates Dimension of X. $W_{m}^{\left(i\right)}$ and $W_{c}^{\left(i\right)}$ are the mean weight and covariance weight of the ith sigma point, respectively. In the above equation, λ is a scaling parameter used to reduce the total prediction error. α, κ, and β are the parameters to be selected, α is a small positive constant ($10^{-4}\leq \alpha \leq 1$); $\kappa =3-N_{x}$ has to ensure that the matrix $(N_{x}+\lambda )P_{XX}$ is semi-positive definite; β is related to the form of the distribution of X, β=2 is optimal for a Gaussian distribution.

Finally, the mean and covariance matrix after the nonlinear mapping is calculated as
(8)$$\bar{Y}\approx \sum_{i=0}^{2N_{x}}W_{m}^{\left(i\right)}Y^{\left(i\right)}$$
(9)$$P_{YY}=\sum_{i=0}^{2N_{x}}W_{c}^{\left(i\right)}[Y^{\left(i\right)}-\bar{Y}]{\left[Y^{\left(i\right)}-\bar{Y}\right]}^{T}$$

#### 2.1.2. UKF for State Estimation

Standard UKF is based on the UT, which takes the transformed mean and covariance and is used to calculate the optimal gain array. Generally, UKF performs two UTs, using the state equation and measurement equation for nonlinear transfer, which correspond to the time update phase and measurement update phase of UKF, respectively. For the standard nonlinear system, the steps of UKF are as follows [4,25].

Firstly, in order to make the state estimation value always unbiased in the filtering process, i.e., $E[{\hat{x}}_{k}]=E[x_{k}]$, the initial condition of the UKF can be given by
(10)$${\hat{x}}_{0}=E[x_{0}]$$
(11)$$P_{0}=E[(x_{0}-{\hat{x}}_{0}){\left(x_{0}-{\hat{x}}_{0}\right)}^{T}]$$
where ${\hat{x}}_{0}$ is the initial state vector of UKF; $P_{0}$ is the initial covariance matrix; and $x_{0}$ is the initial true state vector.

In the time update stage, according to the initial condition, the $2N_{x}+1$ sigma points $x_{k-1|k-1}^{\left(i\right)}$ can be obtained as
(12)$$\chi_{k-1|k-1}^{\left(i\right)}=\left\{ \begin{array}{l} {\hat{x}}_{k-1},i=0\\ {\hat{x}}_{k-1}+p_{k-1,i},i=1\sim 2N_{x} \end{array}\right.\;\sqrt{(N_{x}+\lambda )P_{k-1}}=[p_{1,k-1},p_{2,k-1},\cdots ,p_{N_{x},k-1}]\;-\sqrt{(N_{x}+\lambda )P_{k-1}}=[p_{N_{x}+1,k-1},p_{N_{x}+2,k-1},\cdots ,p_{2N_{x},k-1}]$$
where $\chi_{k-1|k-1}^{\left(i\right)}$ is the ith sigma point of ${\hat{x}}_{k-1}$ at time step k, $p_{k-1,i}$ is the ith column of $\sqrt{(N_{x}+\lambda )P_{k-1}}$ or $-\sqrt{(N_{x}+\lambda )P_{k-1}}$, which donate the ith column of the square root of the lower triangular decomposition of $(N_{x}+\lambda )P_{k-1}$.

Next, the nonlinear transfer is performed on the sigma points and the priori estimation is calculated as
(13)$$\chi_{k|k-1}^{\left(i\right)}=f(\chi_{k-1|k-1}^{\left(i\right)},u_{k-1},\theta )$$
(14)$${\hat{x}}_{k|k-1}=\sum_{i=0}^{2N_{x}}W_{m}^{{}^{\left(i\right)}}\chi_{k|k-1}^{\left(i\right)}$$
(15)$$P_{k|k-1}=\sum_{i=0}^{2N_{x}}W_{c}^{\left(i\right)}[{\hat{x}}_{k|k-1}-\chi_{k|k-1}^{\left(i\right)}]{\left[{\hat{x}}_{k|k-1}-\chi_{k|k-1}^{\left(i\right)}\right]}^{T}+Q$$
where $\chi_{k|k-1}^{\left(i\right)}$ is the one-step prediction of sigma points, ${\hat{x}}_{k|k-1}$ is the priori estimation of the state vector, and $P_{k|k-1}$ is the priori estimation of the covariance matrix.

In the measurement update stage, a new set of sigma points is generated as
(16)$$\chi_{k|k-1}^{*(i)}=\left\{ \begin{array}{l} {\hat{x}}_{k|k-1},i=0\\ {\hat{x}}_{k|k-1}+p_{k|k-1,i},i=1\sim 2N_{x} \end{array}\right.\;\sqrt{(N_{x}+\lambda )P_{k|k-1}}=[p_{1,k|k-1},p_{2,k|k-1},\cdots ,p_{N_{x},k|k-1}]\;-\sqrt{(N_{x}+\lambda )P_{k|k-1}}=[p_{N_{x}+1,k|k-1},p_{N_{x}+2,k|k-1},\cdots ,p_{2N_{x},k|k-1}]$$
where $\chi_{k|k-1}^{*(i)}$ is the ith new sigma point of ${\hat{x}}_{k-1}$ at time step k, $p_{k|k-1,i}$ is the ith column of $\sqrt{(N_{x}+\lambda )P_{k|k-1}}$ or $-\sqrt{(N_{x}+\lambda )P_{k|k-1}}$, which donate the ith column of the square root of the lower triangular decomposition of $(N_{x}+\lambda )P_{k|k-1}$.

As in the time update stage, a nonlinear transfer of measurement equation is performed on the new sigma points and the priori estimation of measurement is calculated as
(17)$$z_{k|k-1}^{\left(i\right)}=h(\chi_{k|k-1}^{*(i)},u_{k},\theta )$$
(18)$${\hat{z}}_{k|k-1}=\sum_{i=0}^{2N_{x}}W_{m}^{{}^{\left(i\right)}}z_{k|k-1}^{\left(i\right)}$$
where $z_{k|k-1}^{\left(i\right)}$ is the one-step prediction of measurements of sigma points, ${\hat{z}}_{k|k-1}$ is the priori estimation of the measurement.

The covariance $P_{zz,k}$ and $P_{xz,k}$ can be obtained by the weighted summation of the sigma point set as
(19)$$P_{zz,k}=\sum_{i=0}^{2N_{x}}W_{c}^{{}^{\left(i\right)}}[z_{k|k-1}^{\left(i\right)}-{\hat{z}}_{k|k-1}]{\left[z_{k|k-1}^{\left(i\right)}-{\hat{z}}_{k|k-1}\right]}^{T}+R$$
(20)$$P_{xz,k}=\sum_{i=0}^{2N_{x}}W_{c}^{{}^{\left(i\right)}}[\chi_{k|k-1}^{\left(i\right)}-{\hat{x}}_{k|k-1}]{\left[z_{k|k-1}^{\left(i\right)}-{\hat{z}}_{k|k-1}\right]}^{T}$$

Then, the Kalman gain matrix $K_{k}$ calculation is obtained as
(21)$$K_{k}=P_{xz,k}P_{zz,k}^{-1}$$

Finally, the estimated state and the covariance can be calculated as
(22)$${\hat{x}}_{k}={\hat{x}}_{k|k-1}+K_{k}[z_{k}-{\hat{z}}_{k|k-1}]$$
(23)$$P_{k}=P_{k|k-1}-K_{k}P_{zz,k}K_{k}^{T}$$

According to the estimated state and measurement equation, measurement is reconciled as
(24)$${\hat{z}}_{k}=h({\hat{x}}_{k},u_{k},\theta )$$

The above process is the standard UKF for state estimation, which does not take into account the time-varying parameters in the system and is only applicable when the parameters are known.

### 2.2. UKF-Based SPE

Considering the problem of system parameter estimation, the complex system will contain several uncertain parameters, and assuming that the parameters are time-varying, define the time-varying parameter vector $\theta_{k}$ at time step k, which is a random vector with zero mean. Therefore, SPE can be achieved by considering $\theta_{k}$ as an extension of the state space, which is estimated together with the state vector [29].
(25)$${x^{\prime }}_{k}=\left[\begin{array}{l} x_{k}\\ \theta_{k} \end{array}\right]$$
where ${x^{\prime }}_{k}$ is augmented state vector, which considers the parameters as part of the state.

The nonlinear dynamic system model can be rewritten with the augmented state $x^{\prime }$ as
(26)$$\begin{array}{l} {x^{\prime }}_{k}=f^{\prime }({x^{\prime }}_{k-1},u_{k-1})+{w^{\prime }}_{k-1}\\ \;=\left[\begin{array}{l} f(x_{k-1},u_{k-1},\theta_{k-1})\\ f_{\theta }(\theta_{k-1}) \end{array}\right]+\left[\begin{array}{l} w_{k-1}\\ \varsigma_{k-1} \end{array}\right] \end{array}$$
(27)$$\begin{array}{l} z_{k}=h({x^{\prime }}_{k},u_{k})+v_{k}\\ \;=h(x_{k},u_{k},\theta_{k})+v_{k} \end{array}$$
where $f^{\prime }(\cdot )=\left[\begin{array}{c} f\\ f_{\theta } \end{array}\right]$ is augmented state equation and ${w^{\prime }}_{k-1}=\left[\begin{array}{l} w_{k-1}\\ \varsigma_{k-1} \end{array}\right]$ is augmented process noise vector with covariance matrix $Q^{\prime }$.

The state equation $f_{\theta }$ for parameter vector can be expressed as
(28)$$\theta_{k}=f_{\theta }(\theta_{k-1})+\varsigma_{k-1}=\theta_{k-1}+\varsigma_{k-1}$$
where $\varsigma_{k-1}\sim G\left(0,S\right)$ is the white noise with zero mean satisfying Gaussian distribution, S is covariance matrix of parameters, the value of which is assumed to be small. Q combined with S yields the augmented covariance matrix $Q^{\prime }$.
(29)$$Q^{\prime }=\left[\begin{array}{cc} Q & 0\\ 0 & S \end{array}\right]$$

Therefore, standard UKF can also be extended for the application of SPE. The process of UKF-based SPE is shown in Figure 2 and its algorithm is derived as follows.
(30)$${\chi^{\prime }}_{k-1|k-1}^{\left(i\right)}=\left\{ \begin{array}{l} {{\hat{x}}^{\prime }}_{k-1},i=0\\ {{\hat{x}}^{\prime }}_{k-1}+{p^{\prime }}_{k-1,i},i=1\sim 2(N_{x}+N_{\theta }) \end{array}\right.\;\sqrt{(N_{x}+\lambda ){P^{\prime }}_{k-1}}=[{p^{\prime }}_{1,k-1},\cdots ,{p^{\prime }}_{N_{x},k-1},{p^{\prime }}_{N_{x}+1,k-1},\cdots ,{p^{\prime }}_{N_{x}+N_{\theta },k-1}]\;-\sqrt{(N_{x}+\lambda ){P^{\prime }}_{k-1}}=[{p^{\prime }}_{N_{x}+N_{\theta }+1,k-1},\cdots ,{p^{\prime }}_{2N_{x}+N_{\theta },k-1},{p^{\prime }}_{2N_{x}+N_{\theta }+1,k-1},\cdots ,{p^{\prime }}_{2(N_{x}+N_{\theta }),k-1}]$$
(31)$${\chi^{\prime }}_{k|k-1}^{\left(i\right)}=f^{\prime }({\chi^{\prime }}_{k-1|k-1}^{\left(i\right)},u_{k-1})$$
(32)$${{\hat{x}}^{\prime }}_{k|k-1}=\sum_{i=0}^{2(N_{x}+N_{\theta })}W_{m}^{{}^{\left(i\right)}}{\chi^{\prime }}_{k|k-1}^{\left(i\right)}$$
(33)$${P^{\prime }}_{k|k-1}=\sum_{i=0}^{2(N_{x}+N_{\theta })}W_{c}^{\left(i\right)}[{{\hat{x}}^{\prime }}_{k|k-1}-{\chi^{\prime }}_{k|k-1}^{\left(i\right)}]{\left[{{\hat{x}}^{\prime }}_{k|k-1}-{\chi^{\prime }}_{k|k-1}^{\left(i\right)}\right]}^{T}+Q^{\prime }$$
(34)$${\chi^{\prime }}_{k|k-1}^{*(i)}=\left\{ \begin{array}{l} {{\hat{x}}^{\prime }}_{k|k-1},i=0\\ {{\hat{x}}^{\prime }}_{k|k-1}+{p^{\prime }}_{k|k-1,i},i=1\sim 2(N_{x}+N_{\theta }) \end{array}\right.\;\sqrt{(N_{x}+\lambda ){P^{\prime }}_{k|k-1}}=[{p^{\prime }}_{1,k|k-1},\cdots ,{p^{\prime }}_{N_{x},k|k-1},{p^{\prime }}_{N_{x}+1,k|k-1},\cdots ,{p^{\prime }}_{N_{x}+N_{\theta },k|k-1}]\;-\sqrt{(N_{x}+\lambda ){P^{\prime }}_{k|k-1}}=[{p^{\prime }}_{N_{x}+N_{\theta }+1,k|k-1},\cdots ,{p^{\prime }}_{2N_{x}+N_{\theta },k|k-1},{p^{\prime }}_{2N_{x}+N_{\theta }+1,k|k-1},\cdots ,{p^{\prime }}_{2(N_{x}+N_{\theta }),k|k-1}]$$
(35)$$z_{k|k-1}^{\left(i\right)}=h({\chi^{\prime }}_{k|k-1}^{*(i)},u_{k})$$
(36)$${\hat{z}}_{k|k-1}=\sum_{i=0}^{2(N_{x}+N_{\theta })}W_{m}^{{}^{\left(i\right)}}z_{k|k-1}^{\left(i\right)}$$
(37)$$P_{zz,k}=\sum_{i=0}^{2(N_{x}+N_{\theta })}W_{c}^{{}^{\left(i\right)}}[z_{k|k-1}^{\left(i\right)}-{\hat{z}}_{k|k-1}]{\left[z_{k|k-1}^{\left(i\right)}-{\hat{z}}_{k|k-1}\right]}^{T}+R$$
(38)$${P^{\prime }}_{xz,k}=\sum_{i=0}^{2(N_{x}+N_{\theta })}W_{c}^{{}^{\left(i\right)}}[{\chi^{\prime }}_{k|k-1}^{\left(i\right)}-{{\hat{x}}^{\prime }}_{k|k-1}]{\left[z_{k|k-1}^{\left(i\right)}-{\hat{z}}_{k|k-1}\right]}^{T}$$
(39)$${K^{\prime }}_{k}={P^{\prime }}_{xz,k}P_{zz,k}^{-1}$$
(40)$${P^{\prime }}_{k}={P^{\prime }}_{k|k-1}-{K^{\prime }}_{k}P_{zz,k}{K^{\prime }}_{k}^{T}$$

The estimated states and parameters are calculated as
(41)$$\left[\begin{array}{c} {\hat{x}}_{k}\\ {\hat{\theta }}_{k} \end{array}\right]={{\hat{x}}^{\prime }}_{k|k-1}+{K^{\prime }}_{k}[z_{k}-{\hat{z}}_{k|k-1}]$$

Finally, the measurements are reconciled as
(42)$$\left[\begin{array}{c} {\hat{x}}_{k}\\ {\hat{\theta }}_{k} \end{array}\right]={{\hat{x}}^{\prime }}_{k|k-1}+{K^{\prime }}_{k}[z_{k}-{\hat{z}}_{k|k-1}]$$

This strategy is known as artificial evolution, which treats fixed model parameters θ as stochastic time-varying model parameters $\theta_{k}$ [29]. The ability of SPE to track the variation of parameters depends heavily on the covariance matrix S. In the practical case, the parameters are fixed and S should be zero, but then UKF will lose the ability of parameter estimation and the estimated parameters will remain time-invariant. If S is large, the stochastic wander implies an increase in covariance as the number of iterations increases, which will cause the parameters $\theta_{k}$ to gradually deviate from the fixed parameters θ. Therefore, S is usually set to a matrix of small values.

## 3. UKF-Based Robust SPE for System with Variable Parameters

UKF-SPE fixes the variance of each system parameter to a small value, and when the parameters are fixed, UKF-SPE gives good estimates. However, the values of parameters can change when the operating conditions switch or when certain faults occur. Due to the limitations of traditional UKF-SPE, changes in the parameters cannot be tracked quickly. Since the covariance matrix S in UKF-SPE is small and fixed, the parameters change slowly during the UKF-SPE tracking iterations, which will produce large estimation errors. This section proposes a reliable method to solve the problem.

At the time step k, the parameters of the process model change, which will result in a corresponding change in the estimated parameters. If the parameters are fixed, the estimated parameters will not change accordingly and converge to a fixed value. Parameter testing based on the trend of parameter changes can be used to identify parameter changes. Once changes in parameters are detected, a gradient descent method is used to speed up the tracking of the parameter, and the learning rate is adaptively adjusted to stabilize the estimated parameters.

### 3.1. Parameters Test

Assuming that UKF-SPE correctly estimates the augmented state, the parameters and states estimated by Equation (41) can be split from this vector. If the parameters do not change, the values of estimated parameters satisfy normal distribution in the vicinity of the correct parameters. Once the system has switched and the parameters have changed, the estimated parameters will deviate from their original values with certain rules. Therefore, based on the corresponding characteristics of parameters, for diagnosing whether the estimated parameters are true or not, a moving window containing information about the estimated parameters in the time series is used to calculate the mean ${\bar{\theta }}_{n,k}$ and variance $s_{n,k}^{2}$ of each parameter in the window, as shown in Figure 3.
(43)$${\bar{\theta }}_{n,k}=\frac{1}{W}\sum_{i=0}^{W-1}{\hat{\theta }}_{n,k-i}$$
(44)$$s_{n,k}^{2}=\frac{1}{W-1}\sum_{i=0}^{W-1}{\left({\hat{\theta }}_{n,k-i}-{\bar{\theta }}_{n,k}\right)}^{2}$$
where W is the size of the move window. Regarding the choice of W, W is a hyperparameter, which is usually obtained according to the empirical method. W needs to be set to an appropriate size. If W is too small, the parameters within the window will have a certain size of variance due to random noise. Conversely, if W is too large, the window contains too much historical information, which can lead to misclassification of the correct parameters.

In order to distinguish whether the parameters are true or not based on their means and variances, statistical methods based on hypothesis testing are used. Firstly, the null hypothesis ($H_{0}$) and the alternative hypothesis ($H_{1}$) can be defined as
(45)$$\begin{array}{l} H_{0}:{\hat{\theta }}_{i,k}=\theta_{i}\\ H_{1}:{\hat{\theta }}_{i,k}\neq \theta_{i} \end{array}$$
where ${\hat{\theta }}_{i,k}$ and $\theta_{i}$ is the ith estimated parameter and true parameter, respectively.

Due to the parameter noise $\varsigma_{k}$ is related to S, the test statistic for the $\chi^{2}$ distribution can be established as
(46)$$\chi^{2}(W-1)=\frac{\left(W-1\right)}{S_{n,n}}s_{n,k}^{2}$$

Based on the quantified variance set for parameters, statistics determined $\chi_{\alpha }^{2}(W-1)$ by the significance level α can be set and the probability event can be described as
(47)$$P\left\{s_{n,k}^{2}\geq \frac{\chi_{\alpha }^{2}(W-1)}{W-1}S_{n,n}\right\}=P\left\{s_{n,k}^{2}\geq s_{th}^{2}\right\}=\alpha$$

Based on the threshold $s_{th}^{2}$, $s_{n,k}^{2}$ can be tested with the equivalent null hypothesis ($H_{0}$) and alternative hypothesis ($H_{1}$):(48)$$\begin{array}{l} H_{0}:{\hat{\theta }}_{i,k}=\theta_{i}\propto s_{n,k}^{2}\leq s_{th}^{2}\\ H_{1}:{\hat{\theta }}_{i,k}\neq \theta_{i}\propto s_{n,k}^{2}>s_{th}^{2} \end{array}$$

Since the parameters estimated by UKF-SPE fluctuate within a small range of the correct parameters, the variance of the latest correct parameters is much smaller than the variance of the incorrect parameters, which is close to zero. If the variance of the estimated parameters in the moving window is below the defined threshold ($s_{th}^{2}$), then the estimated parameters ${\hat{\theta }}_{k}$ is identified as the true parameters; otherwise, they are the false parameters. Once an estimated parameter is judged to be false, then the RMSProp gradient descent method is used to modify the parameters.

### 3.2. RMSProp Gradient Descent

Once the system has switched, the parameter test results will judge that the parameters have changed and the estimated parameters will deviate from their original values, showing a trend towards the true parameters, which can be defined as gradient. With the gradient descent method, the information of the loss function $E_{k}$ is used to calculate the gradient [8], and the parameters are corrected along the direction of the gradient. The gradient descent method is to modify the parameters by iterations to minimize the loss. Assuming that the measurement is desired, the loss function is set to squared error, expressed as
(49)$$E_{k}=\frac{1}{2}{\left(z_{k}-{\hat{z}}_{k}\right)}^{T}(z_{k}-{\hat{z}}_{k})$$

According to the rule of back propagation, the gradient can be calculated as
(50)$$\Gamma_{k-1}=\frac{\partial E_{k}}{\partial \theta_{k-1}}=-(z_{k}-{\hat{z}}_{k})\frac{\partial h}{\partial \theta_{k-1}}$$

The observation error $r_{k}$ can be obtained by Equation (27).
(51)$$r_{k}=z_{k}-{\hat{z}}_{k}=h(x_{k},u_{k},\theta )+v_{k}-h({\hat{x}}_{k},u_{k},{\hat{\theta }}_{k})$$

Equation (51) can be approximated as a first-order Taylor expansion as follows
(52)$$r_{k}=z_{k}-{\hat{z}}_{k}\approx \frac{\partial h}{\partial \theta_{k}}({\hat{\theta }}_{k}-{\hat{\theta }}_{k-1})$$

The gradient $\Gamma_{k-1}$ is approximated as
(53)$$\Gamma_{k-1}\approx -\frac{\partial h}{\partial \theta_{k}}({\hat{\theta }}_{k}-{\hat{\theta }}_{k-1})\frac{\partial h}{\partial \theta_{k-1}}=-D_{k}\frac{\left({\hat{\theta }}_{k}-{\hat{\theta }}_{k-1}\right)}{T}$$
where $D_{k}=T\frac{\partial h}{\partial \theta_{k}}\frac{\partial h}{\partial \theta_{k-1}}$, T is the interval time between two samples. It can be found that the gradient is related to the trend of the parameters. The gradient is also a modification quantity for the parameters, assuming that the learning rate is η, and the parameters can be modified as
(54)$${\hat{\theta }}_{k-1}^{*}={\hat{\theta }}_{k-1}-\eta \Gamma_{k-1}$$
where ${\hat{\theta }}_{k-1}^{*}$ is the vector of modified parameters, η is a fixed constant which will lead to large fluctuations of parameter estimation results. The ideal learning rate is set large at the beginning, with a fast convergence rate, and then decays slowly to ensure stable arrival at the optimal point, as shown in Figure 4. η can be turned into a time-varying learning rate vector $\eta_{k-1}$. The learning rate adaptive algorithm often uses RMSProp, which uses exponentially decaying averaged gradients to discard past historical information, allowing it to converge quickly after finding the gradient minimum, solving the problem of large oscillations in optimization.

RMSProp needs to calculate the cumulative squared gradient as
(55)$$\lambda_{k-1}=\rho \lambda_{k-2}+(1-\rho )\Gamma_{k-1}⊙\Gamma_{k-1}$$
where ρ∈[0,1) is the rate of reduction. The learning rate of each parameter is then updated to
(56)$$\eta_{i,k-1}=\frac{\varepsilon }{\sqrt{\delta +\lambda_{i,k-1}}}$$
where ε is the global learning rate, δ is a small constant that makes the value stable when divided by a decimal, usually set to 10^−6^. $\eta_{i,k-1}$ A is the learning rate of the ith parameter, RMSProp chooses a different learning rate for each parameter. Equation (54) can be rewritten as
(57)$${\hat{\theta }}_{k-1}^{*}={\hat{\theta }}_{k-1}-\eta_{k-1}⊙\Gamma_{k-1}$$

Finally, with the modified parameter vector ${\hat{\theta }}_{k-1}^{*}$, the UKF-SPE is re-conducted to obtain new ${\hat{x}}_{k}$ and ${\hat{\theta }}_{k}$, the measurements are also reconciled to new ${\hat{z}}_{k}$.

The proposed UKF-based robust SPE considers the effect of parameter changes and is applicable to nonlinear dynamic systems with variable parameters. When the system is switched, or a failure occurs, the system parameters change, and the parameter testing based on hypothesis testing and moving windows is introduced to detect whether the system parameters have changed. Based on the detection results, the RMSProp gradient descent method that adaptively modifies the learning rate is used to fast track the changed parameters. The proposed UKF-based robust SPE algorithm is shown as follows (Algorithm 1).
**Algorithm****1:** UKF-based robust SPE**Input:** x^k−1; θ^k−1; zk**Begin:****Step 1.** Extend the state vector to obtain x′k−1.**Step 2.** Conduct UKF-SPE based on x′k−1 to estimate x^k, θ^k, and reconcile z^k.**Step 3.** Conduct parameters test:1. Calculate the mean θ¯n,k and variance sn,k2 of each estimated parameter in the window using Equations (43) and (44);2. Performing hypothesis testing;   For *n* = 1: Nθ     If null hypothesis H0 is accepted, θ^i,k=θi, skip the following steps to output     Else θ^i,k≠θi, proceed to the next step**Step 4.** Conduct RMSProp gradient descent:1. Calculate loss Ek using Equation (49);2. Calculate the approximate gradient Γk−1 using Equation (53);   3. Calculate the learning rate ηi,k−1 for each parameter using Equation (56);4. Modify parameters using Equation (57);**Step 5.** Re-conduct UKF-SPE based on the modified parameters to estimate x^k, θ^k, and reconcile z^k.**End****Output:** x^k; θ^k; z^k

## 4. Case Studies

To verify the effectiveness of the proposed UKF-RSPE, the UKF-RSPE was applied to a typical nonlinear dynamic process system and the free radical polymerization of styrene. All case studies were performed by using MATLAB 2016a.

### 4.1. Case Study 1: Typical Nonlinear Dynamic System

In order to simply show the effect of the proposed UKF-RSPE, the proposed UKF-RSPE is applied to a typical nonlinear dynamic process system, which is commonly used in filter testing [29,37]. Its mathematical model can be expressed as
(58)$$x_{k}=\frac{x_{k-1}}{2}+\frac{\theta x_{k-1}}{1+x_{k-1}^{2}}+8\cos(1.2k)+w_{x,k-1}$$
(59)$$z_{k}=\frac{1}{20}x_{k}^{2}+v_{z,k-1}$$
where $v_{k-1}$ and $v_{z,k-1}$ are the process noise and the measurement noise with standard deviations of 0.1, respectively. θ is the model parameter. Under normal conditions, the value of the parameter θ is set to be 25. If a fault occurs, the parameter changes to be 12.5. To keep track of the changed parameter, the state vector and the state equations are expanded to be
(60)$$x_{k}=\;\left[\begin{array}{l} x_{k}\\ \;\theta_{k}\; \end{array}\right]=\left[\begin{array}{l} \frac{x_{k-1}}{2}+\frac{\theta_{k-1}x_{k-1}}{1+x_{k-1}^{2}}+8\cos(1.2k)\\ \theta_{k-1} \end{array}\right]+\left[\begin{array}{l} w_{x,k-1}\\ \;\varsigma_{\theta ,k-1} \end{array}\right]$$
where $\varsigma_{\theta ,k-1}$ is the parameter noise with the standard deviation of 0.01.

In this simulation, W is set to be 5, α is set to be 0.05. The simulation time step is set to be 500 and the value of parameter changes at time step 200. The initial state is set to be x(0)=[0,25]. The comparison results of SPE are shown in Figure 5, Figure 6 and Figure 7. The comparison of the estimated results of the state in Figure 5 shows that the distribution of system state changes at time step 200. Since the random noise in the simulation is small, it can be clearly seen that the state error of the traditional UKF-SPE estimation increases significantly after the fault occurs in Figure 5b. Compared with the traditional UKF-SPE, the proposed UKF-RSPE can achieve more accurate state estimation. In Figure 6, the comparison of the estimated results of parameter directly shows that the parameter changes from 25 to 12.5. The traditional UKF-SPE cannot track the true parameter quickly, while UKF-RSPE can track the parameters quickly and accurately and reduce the error of parameter estimation. Due to the presence of the learning rate in UKF-RSPE, there is a small fluctuation after the estimated parameter reaches the true value. With the learning rate continuously updating, the parameter is finally stabilized around the true value. Finally, the measurements are reconciled according to the estimated states and parameters. The comparison of the reconciled results of measurements shows that the UKF-RSPE can reconcile measurements, and the measurement error of the UKF-RSPE is smaller than that of UKF-SPE, as shown in Figure 7.

To further quantify the effect of UKF-RSPE, the statistical information of estimated results is listed in Table 1. Compared with the traditional UKF-SPE, the MSE of state estimation is reduced from 20.4359 to 0.8509; and the MSE of parameter estimation is reduced from 26.4655 to 1.3346. The MSE of the measurement data is 0.0106; due to the failure, the traditional UKF-SPE cannot accurately estimate the status, resulting in the MSE of measurements increasing to be 4.3588, but the UKF-RSPE can reduce the MSE to be 0.3375. Therefore, the proposed UKF-RSPE can be applied to the general nonlinear dynamic system with variable parameter and achieve more accurate state and parameter estimation.

### 4.2. Case Study 2: Free Radical Polymerization of Styrene

To verify the effectiveness of the proposed UKF-RSPE, the UKF-RSPE was applied to the free radical polymerization of styrene. This paper uses the mathematical model of dynamics of polymerization in a jacketed continuous stirred tank reactor proposed by Schmidt and Ray [5]. The reaction mechanism of free radical polymerization of styrene is divided into decomposition reaction of initiator, initiator reaction to form monomer radical, chain propagation reaction, and termination reaction. First of all, the reaction mechanisms start with the decomposition reaction of an initiator such as azodiisobutyronitrile (AIBN) [38] which can be represented as
(61)$$I\overset{k_{d,0},E_{d}}{\to }2R^{*}\left(\mathrm{Decomposition}\;\mathrm{reaction}\right)$$
where I is the initiator involved in the decomposition reaction that produces the radical $R^{*}$. $k_{d,0}$ and $E_{d}$ are the specific reaction rate constant and energy of activation of the decomposition reaction.

Hereafter, the radical $R^{*}$ reacts with the monomer M in the initiation reaction to produce $P_{1}$ which is a live polymer of the unit chain length.
(62)$$R^{*}+M\overset{k_{i,0},E_{i}}{\to }P_{1}\left(\mathrm{Initiator}\;\mathrm{reaction}\right)$$
where $k_{i,0}$ and $E_{i}$ are the specific reaction rate constant and energy of activation of the initiation reaction.

The propagation reaction for $P_{n}$, alive polymer of n units of monomer to produce $P_{n+1}$, starting with $P_{1}$ is given by
(63)$$P_{n}+M\overset{k_{p,0},E_{p}}{\to }P_{n+1}\left(\mathrm{Propagation}\;\mathrm{reaction}\right)$$
where $k_{p,0}$ and $E_{p}$ are the corresponding specific reaction rate constant and energy of activation of the propagation reaction

Afterwards, the polymer product is obtained from the dead polymer product $M_{n+m}$ with chain lengths of n+m units of monomer formed by the termination reaction.
(64)$$P_{n}+P_{m}\overset{k_{t,0},E_{t}}{\to }M_{n+m}\left(\mathrm{Termination}\;\mathrm{reaction}\right)$$
where $k_{t,0}$ and $E_{t}$ are the corresponding specific reaction rate constant and energy of activation of the termination reaction. The above reaction conditions are listed in Table 2.

The feed and output flow rates and the feed solvent concentration do not change with time throughout the reaction. It is assumed that the feed is equal to the output and there is no solvent build-up in the reactor.

The mechanistic model describes the actual dynamic process of free radical polymerization of styrene, in which the products have distributional properties, such as the concentrations of the polymers. The estimation of the distribution output is critical as it significantly affects the product quality and process efficiency. The polymer concentration is considered as the process state, and the output of this process is the distribution of polystyrene concentration as a function of polystyrene chain length at the reactor outlet. Simulations of real scenarios revealed that the estimated error in polystyrene concentration for chain lengths longer than 1000 units of monomer is 10^−4^ or less; therefore, the maximum chain length estimated for this process was set to 1000.

The concentration of the initiator ${\tilde{C}}_{I,k}\;$, concentration of the monomer ${\tilde{C}}_{M,k}\;$ and the temperature of reactor ${\tilde{T}}_{k}$ are taken as the process states. The equations of states in the nonlinear dynamic model are given by
(65)$$x_{k}=\;\left[\begin{array}{l} {\tilde{C}}_{I,k}\;\\ \;{\tilde{C}}_{M,k}\;\\ {\tilde{T}}_{k} \end{array}\right]=\left[\begin{array}{l} f_{1}({\tilde{C}}_{I,k-1}\;,{\tilde{T}}_{k-1})\;\\ f_{2}({\tilde{C}}_{I,k-1}\;,{\tilde{C}}_{M,k-1}\;,{\tilde{T}}_{k-1})\;\\ f_{3}({\tilde{C}}_{I,k-1}\;,{\tilde{C}}_{M,k-1}\;,{\tilde{T}}_{k-1})\; \end{array}\right]+\left[\begin{array}{l} w_{1,k-1}\\ w_{2,k-1}\;\\ w_{3,k-1}\; \end{array}\right]$$
where $f_{1}$, $f_{2}$, and $f_{3}$ are state equations of ${\tilde{C}}_{I,k}\;$, ${\tilde{C}}_{M,k}\;$, and ${\tilde{T}}_{k}$, respectively. $w_{1,k-1}$, $w_{2,k-1}$, and $w_{3,k-1}$ are process noise of ${\tilde{C}}_{I,k}\;$, ${\tilde{C}}_{M,k}\;$, and ${\tilde{T}}_{k}$, respectively. The information of ${\tilde{C}}_{I,k}\;$, ${\tilde{C}}_{M,k}\;$, and ${\tilde{T}}_{k}$ is listed in Table 3. At a steady state set-point reactor temperature $T_{ss}$ of 310 K, the following values are obtained.
(66)$$\begin{array}{l} C_{I,ss}=0.0691\;\mathrm{kmol}\cdot {\mathrm{mol}}^{-3}\\ C_{M,ss}=3.393\;\mathrm{kmol}\cdot {\mathrm{mol}}^{-3} \end{array}$$
where ss represents the steady state. At set-point conditions, the steady-state concentration range of the polymer is zero. The magnitudes of changes in the states ${\tilde{C}}_{I,k}$, ${\tilde{C}}_{M,k}\;$, and ${\tilde{T}}_{k}$ are large, therefore, the states must be redefined in terms of dimensionless variables so that when the system reaches the set-point condition, they are scaled to one.
(67)$${\tilde{C}}_{I}=\frac{C_{I}}{C_{I,ss}}$$
(68)$${\tilde{C}}_{M}=\frac{C_{M}}{C_{M,ss}}$$
(69)$$\tilde{T}=\frac{T}{T_{ss}}$$

The state equations $f_{1}$, $f_{2}$, and $f_{3}$ can be obtained from the differential equations, which, rewritten in terms of scaled dimensionless variables, are listed as follows.
(70)$$\frac{d{\tilde{C}}_{I}}{dt}=-\frac{{\tilde{C}}_{I}}{\tau }-k_{d,0}\exp\left[\frac{\left(-\frac{E_{d}}{RT_{ss}}\right)}{\tilde{T}}\right]{\tilde{C}}_{I}+\left(\frac{F_{i}}{V}\right){\tilde{C}}_{I,i}$$
(71)$$\frac{d{\tilde{C}}_{M}}{dt}=-\frac{{\tilde{C}}_{M}}{\tau }-k_{p,0}\exp\left[\frac{\left(-\frac{E_{p}}{RT_{ss}}\right)}{\tilde{T}}\right]C_{P}{\tilde{C}}_{M}+\left(\frac{F_{m}}{V}\right){\tilde{C}}_{M,m}$$
(72)$$\frac{d\tilde{T}}{dt}=\frac{{\tilde{T}}_{i}-\tilde{T}}{\tau }+\frac{\gamma C_{m,ss}k_{p.0}}{T_{ss}}\exp\left[\frac{\left(-\frac{E_{p}}{RT_{ss}}\right)}{\tilde{T}}\right]{\tilde{C}}_{M}C_{P}+\left(\frac{Q_{ss}}{\rho c_{p}VT_{ss}}\right)\tilde{Q}$$
where the information of parameters in the above equations is listed in Table 2 and Table 4. The total concentration of live polymers $C_{P}$ is expressed in terms of dimensionless variables ${\tilde{C}}_{I}$ and $\tilde{T}$ by
(73)$$\begin{array}{l} C_{P}=\sqrt{\frac{2C_{I}fk_{d}}{k_{t}}}=\sqrt{\frac{2{\tilde{C}}_{I}\;\cdot \;C_{I,ss}fk_{d,0}\exp\left[\frac{\left(\frac{-E_{d}}{RT_{ss}}\right)}{\left(\frac{T}{T_{ss}}\right)}\right]}{k_{t,0}\exp\left[\frac{\left(\frac{-E_{t}}{RT_{ss}}\right)}{\left(\frac{T}{T_{ss}}\right)}\right]}}\;\\ =\sqrt{\frac{2C_{I,ss}fk_{d,0}}{k_{t,0}}}\cdot \sqrt{{\tilde{C}}_{I}\exp\left[\frac{\left(\frac{E_{t}-E_{d}}{RT_{ss}}\right)}{\tilde{T}}\right]} \end{array}$$
where f=0.6 is an initiator efficiency factor which accounts for the fraction of initiator participating in the chain initiation reaction.

Considering the state vector and $C_{P}$, the measurement vector is set to $z_{k}=\;\left[y_{1,k};y_{2,k};y_{3,k}\;;y_{4,k}\right]=\left[{\tilde{C}}_{I,k};{\tilde{C}}_{M,k}\;;{\tilde{T}}_{k}\;;C_{P}\;\right]$, the measurement equations can be expressed as
(74)$$z_{k}=\;\left[\begin{array}{l} y_{1,k}\\ y_{2,k}\\ \;y_{3,k}\;\\ y_{4,k} \end{array}\right]=\left[\begin{array}{l} {\tilde{C}}_{I,k}\\ {\tilde{C}}_{M,k}\;\\ {\tilde{T}}_{k}\;\\ C_{P}\; \end{array}\right]+\left[\begin{array}{l} v_{1,k}\\ v_{2,k}\\ v_{3,k}\\ v_{4,k} \end{array}\right]$$
where $v_{1,k}$, $v_{2,k}$, $v_{2,k}$, and $v_{3,k}$ are the measurement noise with standard deviations of 0.001, 0.001, 0.001, and 1, respectively

To verify the performance of the proposed UKF-based robust SPE, it was applied to the simulations of free radical polymerization of styrene and compared with traditional UKF-SPE. Two independent cases were considered, in which there are different combinations of uncertain parameters that change when failure occurs. Case Study 2.1 has only a single uncertain parameter, while Case Study 2.2 contains multiple uncertain parameters. The other parameters are the same in the two cases. W is set to be 5, α is set to be 0.05. The simulation time step is set to be 800 and the parameters are changed at time step 300.

#### 4.2.1. Case Study 2.1: Robust SPE with Single Parameter

The initiator efficiency factor f is a constant which is assumed to be an uncertain parameter. With this parameter included, the state vector and the state equations are expanded to
(75)$$x_{k}=\;\left[\begin{array}{l} {\tilde{C}}_{I,k}\;\\ \;{\tilde{C}}_{M,k}\;\\ {\tilde{T}}_{k}\\ f \end{array}\right]=\left[\begin{array}{l} f_{1}({\tilde{C}}_{I,k-1}\;,{\tilde{T}}_{k-1})\;\\ f_{2}({\tilde{C}}_{I,k-1}\;,{\tilde{C}}_{M,k-1}\;,{\tilde{T}}_{k-1})\;\\ f_{3}({\tilde{C}}_{I,k-1}\;,{\tilde{C}}_{M,k-1}\;,{\tilde{T}}_{k-1})\;\\ \;f_{\theta }(f) \end{array}\right]+\left[\begin{array}{l} w_{1,k-1}\\ w_{2,k-1}\;\\ w_{3,k-1}\\ \;\varsigma_{f,k-1} \end{array}\right]$$
where $\varsigma_{f,k-1}$ is the parameter noise with the standard deviation of 10^−^^6^, f is 0.6 under normal conditions. When a fault occurs, the parameter f changes to 0.3 at time step 300.

The comparison results of SPE are shown in Figure 8, Figure 9, Figure 10 and Figure 11. The effect of parameter f change on states ${\tilde{C}}_{I,k}$ and ${\tilde{C}}_{M,k}$ is not significant, the parameter f mainly affects the temperature of reactor ${\tilde{T}}_{k}$. Figure 8 shows the comparison of the estimated results of the state ${\tilde{T}}_{k}$, it can be clearly seen that starting from time step 300, estimated state ${\tilde{T}}_{k}$ of the traditional UKF-SPE deviates from the theoretical true state. The UKF-SPE can only slowly track the true state, while the proposed UKF-RSPE can maintain accurate state estimation. In Figure 9, from the comparison of the estimated results of parameter f, it is clear that the traditional UKF-SPE tracks the parameter slowly, while the UKF-RSPE can track the true parameter much faster. Due to the presence of the learning rate in UKF-RSPE, there is a certain fluctuation when the true parameter is first tracked, which leads to misdetection as a false parameter, but with iteration, the learning rate is continuously updated, accompanied by past information being forgotten until the parameter f is finally stabilized as the true parameter through parameter testing. It can also be observed in Figure 9 that the fluctuations appear at the beginning of the parameter estimation due to the adjustment of the covariance $P_{k}$ from the initial unit matrix with iterations. Finally, the measurements are reconciled according to the estimated states and parameter. The comparison of the reconciled results of measurement ${\tilde{T}}_{k}$ is the same as the estimated results of the state ${\tilde{T}}_{k}$ as shown in Figure 10. The proposed UKF-RSPE can reconcile measurements, and the errors of the measurements are smaller than the measurements, indicating that the UKF-RSPE still has the ability to reconcile the data in the event of a fault. The change of f not only affects ${\tilde{T}}_{k}$, but also the measurement of the total concentration of live polymers $C_{p}$. The comparison of the reconciled results of measurement $C_{p}$ is shown in Figure 11; $C_{p}$ undergoes a direct change when s fault occurs indicating that f has a large effect on $C_{p}$. The UKF-RSPE can reconcile the measurements quickly, while the traditional UKF-SPE can only track the measurements slowly. The measurement error of the UKF-RSPE is smaller than the measurement and the measurement of UKF-SPE, as shown in Figure 11b.

To further quantify the effect of UKF-RSPE, the statistical information of estimated results is listed in Table 5. Compared with the traditional UKF-SPE, the mean square error (MSE) of state estimation is reduced from 1.51 × 10^−6^ to 8.67 × 10^−7^; and the MSE of parameter estimation is reduced from 7.09 × 10^−3^ to 2.08 × 10^−4^. The MSE of the measurement data is 2.49 × 10^−1^, and the traditional UKF-SPE not only fails to correct the measurements, but also increases the MSE to 1.03. The UKF-RSPE can reconcile the measurements and reduce the MSE to 8.58 × 10^−2^. Therefore, the proposed UKF-RSPE tracks the parameter change faster and can achieve more accurate state and parameter estimation.

#### 4.2.2. Case Study 2.2: Robust SPE with Multiple Parameters

The reaction rate constant of the propagation reaction $k_{p,0}$ and the termination reaction $k_{t,0}$ are assumed to be uncertain parameters. With the parameters included, the state vector and the state equations are expanded to
(76)$$x_{k}=\;\left[\begin{array}{l} {\tilde{C}}_{I,k}\;\\ \;{\tilde{C}}_{M,k}\;\\ {\tilde{T}}_{k}\\ k_{p,0}\\ k_{t,0} \end{array}\right]=\left[\begin{array}{l} f_{1}({\tilde{C}}_{I,k-1}\;,{\tilde{T}}_{k-1})\;\\ f_{2}({\tilde{C}}_{I,k-1}\;,{\tilde{C}}_{M,k-1}\;,{\tilde{T}}_{k-1})\;\\ f_{3}({\tilde{C}}_{I,k-1}\;,{\tilde{C}}_{M,k-1}\;,{\tilde{T}}_{k-1})\;\\ \;f_{\theta }(k_{p,0})\\ f_{\theta }(k_{t,0}) \end{array}\right]+\left[\begin{array}{l} w_{1,k-1}\\ w_{2,k-1}\;\\ w_{3,k-1}\\ \;\varsigma_{p,k-1}\\ \varsigma_{t,k-1} \end{array}\right]$$
where $\varsigma_{p,k-1}$ and $\varsigma_{t,k-1}$ are the parameter noises with the standard deviation of 10^−6^. Under normal conditions, the parameter values are set to be $k_{p,0}=1.00\times 10^{7}\;\mathrm{kmol}\cdot m^{-3}\cdot s^{-1}$, $k_{t,0}=1.25\times 10^{7}\;\mathrm{kmol}\cdot m^{-3}\cdot s^{-1}$. When a fault occurs, the parameters change to be $k_{p,0}=1.50\times 10^{7}\;\mathrm{kmol}\cdot m^{-3}\cdot s^{-1}$, $k_{t,0}=0.6\times 10^{7}\;\mathrm{kmol}\cdot m^{-3}\cdot s^{-1}$ at time step 300.

The comparison results of SPE are shown in Figure 12, Figure 13, Figure 14, Figure 15 and Figure 16. In this case, the simultaneous change of both parameters has a dramatic effect on the state estimation. In Figure 12, the temperature of the reactor ${\tilde{T}}_{k}$ estimated by traditional UKF-SPE changes abruptly and UKF-SPE takes longer time to track the true state. The proposed UKF-RSPE can rapidly estimate the true state and improve the accuracy of state estimation. Figure 13 and Figure 14 compare the estimated results of parameters $k_{p,0}$ and $k_{t,0}$, respectively. For $k_{t,0}$, UKF-RSPE maintains its excellent performance in the case with single parameter, as shown in Figure 14a. However, for $k_{p,0}$, although UKF-RSPE tracks the true parameter faster compared to UKF-SPE, it still takes more than 150 time steps due to the interplay between multiple parameters when they are estimated simultaneously. There is also some fluctuation after tracking the true parameters until the parameter detection is judged to be true. Fluctuations in the vicinity of the true parameters have small effects on the state estimation and measurement reconciliation, so they can be ignored and the estimated parameters can be considered as true parameters. Figure 15 shows the comparison of the reconciled results of measurement ${\tilde{T}}_{k}$; UKF-RSPE can reconcile the measurements and reduce the errors of measurements, which greatly improves on the accuracy of traditional UKF-SPE. Figure 16 shows the comparison of the reconciled results of measurement $C_{p}$, a sudden change occurred at the time of the fault, both the traditional UKF-SPE and UKF-RSPE are able to reconcile the measurements, the errors of UKF-RSPE are smaller.

To further quantify the effect of UKF-RSPE, the statistical information of estimated results is listed in Table 6. Compared with the traditional UKF-SPE, the MSEs of state estimation and parameter estimation are reduced from 7.17 × 10^−6^ and 4.53 × 10^−2^, respectively, to 9.20 × 10^−7^ and 6.04 × 10^−3^. The MSE of the measurement data is 2.59 × 10^−1^ and the MSE of the measurements reconciled by UKF-RSPE is reduced to 2.35 × 10^−1^. In summary, the proposed UKF-RSPE can track all the parameters changes faster not only in the case with single parameter but also in the case with different combinations of multiple parameters changes. The proposed UKF-RSPE can achieve more accurate state and parameter estimates and has stronger robustness.

The proposed UKF-based robust SPE is not limited to free radical polymerization of styrene, it can be applied to other radical polymerization processes. The free radical polymerization of styrene described in this section is just one case of the application of UKF-based robust SPE. UKF is a model-based approach, and by changing the mathematical model, it can be applied in the case of other radical polymerization processes. However, when the UKF-based robust SPE is applied to other radical polymerization processes, the mathematical models of other radical polymerization processes need to be obtained firstly. In general, the chemical reaction is complicated, and it may be difficult to derive the accurate mathematical model of the chemical reaction. Therefore, a characteristic of UKF is model-driven. Based on being model-driven, it can be used in other radical polymerization processes.

## 5. Conclusions

In this work, the SPE problem of the free radical polymerization of styrene with variable parameters was studied, given that there are uncertain parameters in the mechanistic model. When the reaction conditions are switched, or the reaction process generates faults, the parameters will change. Due to the limitation of traditional UKF-SPE in detecting and fast-tracking changes in model parameters, this paper proposed a UKF-based robust SPE method, which establishes a moving window and introduces a hypothesis-test-based parameter testing criterion within the window to detect whether the parameters have changed. Based on the detection results, the gradient descent method which adaptively modifies the learning rate is used to modify the parameters to accelerate the tracking of the parameters, and to obtain more accurate parameters and states. The proposed UKF-based robust SPE is applied to the simulation study of the free radical polymerization of styrene using the mathematical model of dynamics of polymerization in a jacketed continuous stirred tank reactor. The model is based on the reaction mechanism and describes the dynamics of the actual free radical polymerization of styrene. The experimental results show that the proposed UKF-based robust SPE can quickly track the parameter changes and achieve more accurate state and parameter estimation with stronger robustness.

## Figures and Tables

**Figure 1 polymers-14-00973-f001:**
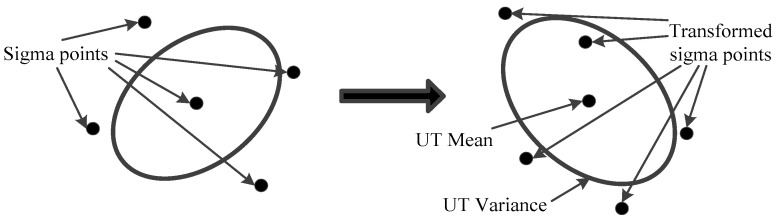
The schematic of unscented transformation.

**Figure 2 polymers-14-00973-f002:**
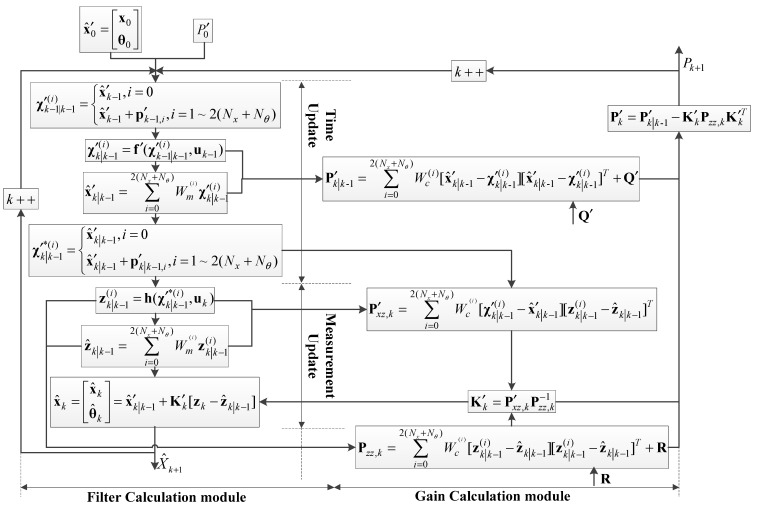
The process of UKF-based SPE.

**Figure 3 polymers-14-00973-f003:**
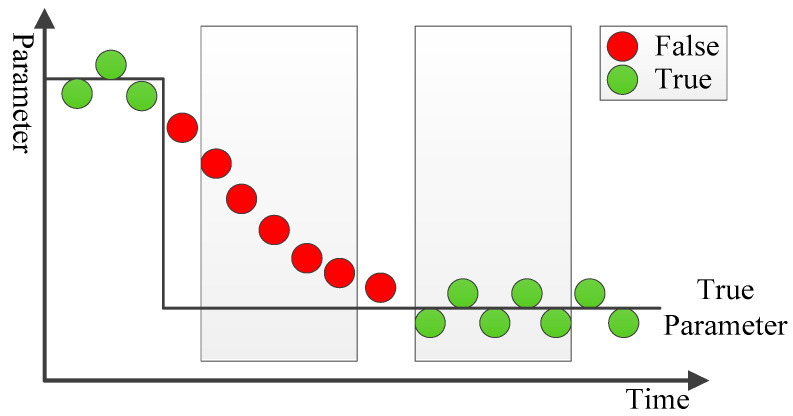
The moving window for parameter testing.

**Figure 4 polymers-14-00973-f004:**
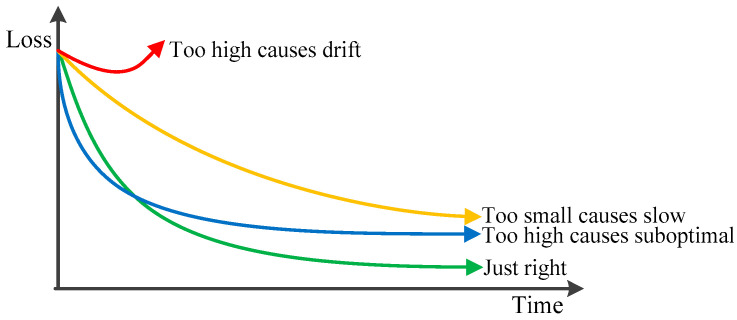
The tracking effect of different learning rates.

**Figure 5 polymers-14-00973-f005:**
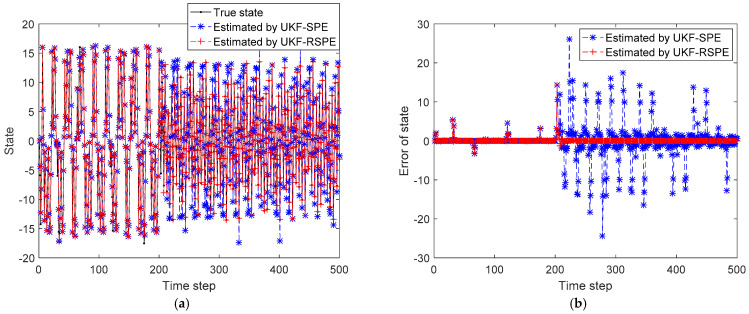
The comparison of the estimated results of the state in Case Study 1: (**a**) estimated state; (**b**) state error.

**Figure 6 polymers-14-00973-f006:**
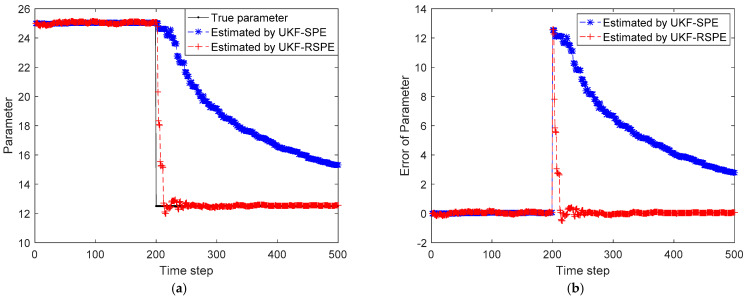
The comparison of the estimated results of parameter in Case Study 1: (**a**) estimated parameter; (**b**) parameter error.

**Figure 7 polymers-14-00973-f007:**
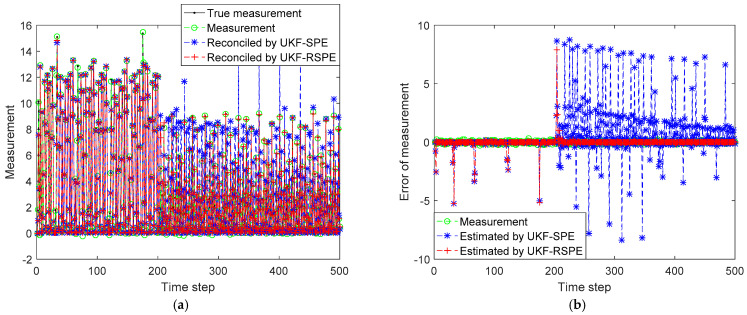
The comparison of the reconciled results of measurement in Case Study 1: (**a**) reconciled measurement; (**b**) measurement error.

**Figure 8 polymers-14-00973-f008:**
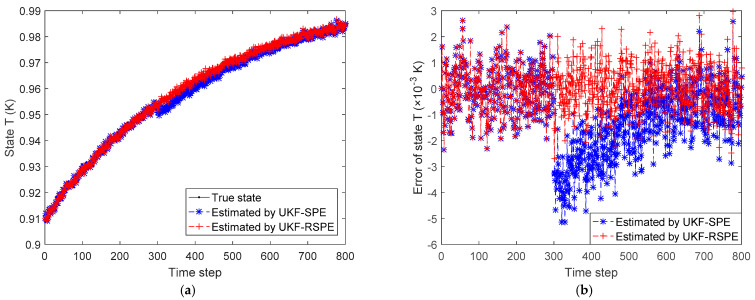
The comparison of the estimated results of the state ${\tilde{T}}_{k}$ in Case Study 2.1: (**a**) estimated state; (**b**) state error.

**Figure 9 polymers-14-00973-f009:**
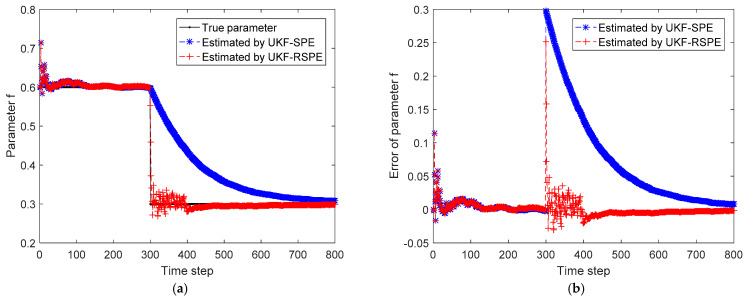
The comparison of the estimated results of parameter f in Case Study 2.1: (**a**) estimated parameter; (**b**) parameter error.

**Figure 10 polymers-14-00973-f010:**
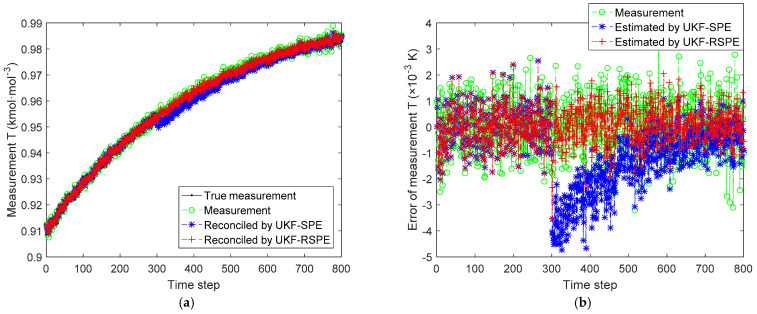
The comparison of the reconciled results of measurement ${\tilde{T}}_{k}$ in Case Study 2.1: (**a**) reconciled measurement; (**b**) measurement error.

**Figure 11 polymers-14-00973-f011:**
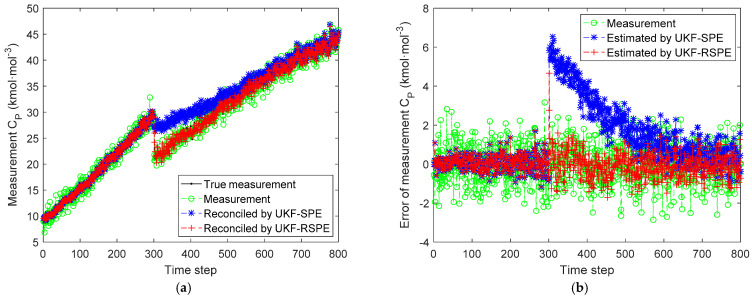
The comparison of the reconciled results of measurement $C_{p}$ in Case Study 2.1: (**a**) reconciled measurement; (**b**) measurement error.

**Figure 12 polymers-14-00973-f012:**
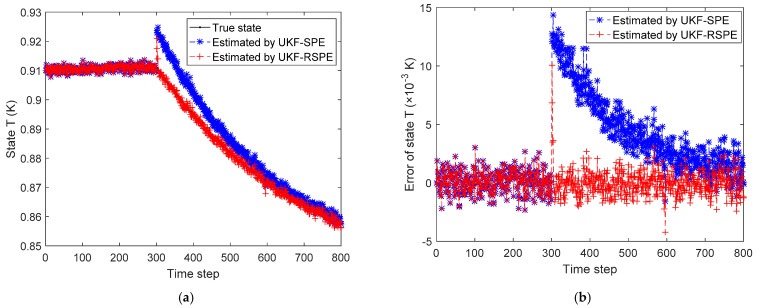
The comparison of the estimated results of the state ${\tilde{T}}_{k}$ in Case Study 2.2: (**a**) estimated state; (**b**) state error.

**Figure 13 polymers-14-00973-f013:**
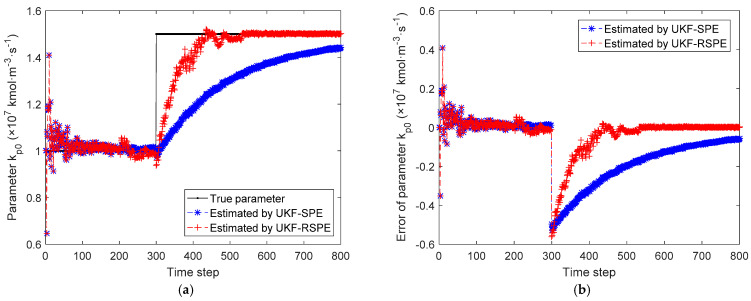
The comparison of the estimated results of parameter $k_{p,0}$ in Case Study 2.2: (**a**) estimated parameter; (**b**) parameter error.

**Figure 14 polymers-14-00973-f014:**
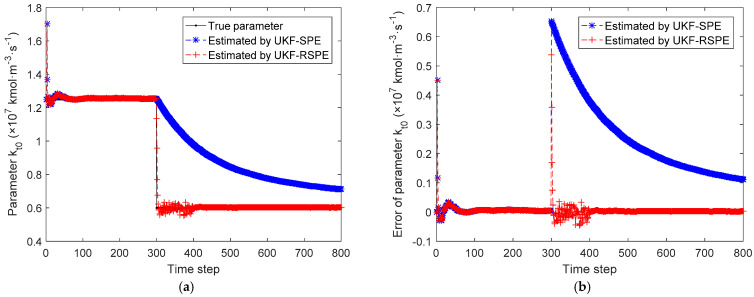
The comparison of the estimated results of parameter $k_{t,0}$ in Case Study 2.2: (**a**) estimated parameter; (**b**) parameter error.

**Figure 15 polymers-14-00973-f015:**
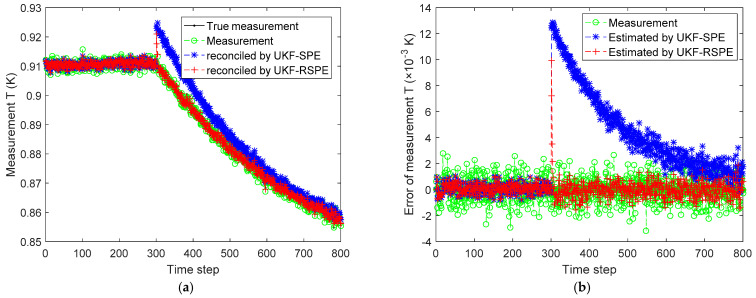
The comparison of the reconciled results of measurement ${\tilde{T}}_{k}$ in Case Study 2.2: (**a**) reconciled measurement; (**b**) measurement error.

**Figure 16 polymers-14-00973-f016:**
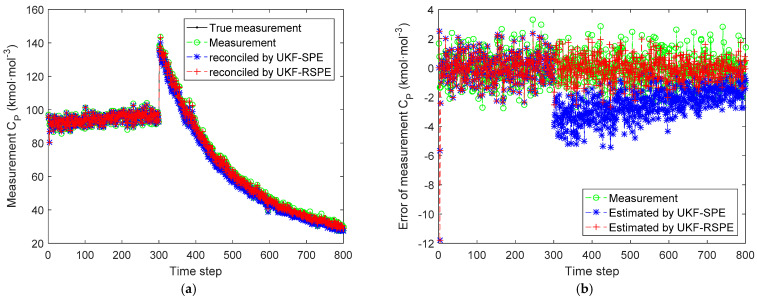
The comparison of the reconciled results of measurement $C_{p}$ in Case Study 2.2: (**a**) reconciled measurement; (**b**) measurement error.

**Table 1 polymers-14-00973-t001:** Statistical information of estimated results in Case Study 1.

Results	MSE	
	UKF-SPE	UKF-RSPE
States	20.4359	0.8509
Parameters	26.4655	1.3346
Measurements (MSE = 0.0106)	4.3588	0.3375

**Table 2 polymers-14-00973-t002:** Information of the reaction conditions in Case Study 2.

Process Parameters	Value
The reaction rate constant of the decomposition reaction $k_{d,0}$	$5.95\times 10^{13}\;\mathrm{kmol}\cdot m^{-3}\cdot s^{-1}$
The reaction rate constant of the propagation reaction $k_{p,0}$	$1.00\times 10^{7}\;\mathrm{kmol}\cdot m^{-3}\cdot s^{-1}$
The reaction rate constant of the termination reaction $k_{t,0}$	$1.25\times 10^{7}\;\mathrm{kmol}\cdot m^{-3}\cdot s^{-1}$
The energy of activation of the decomposition reaction $E_{d}$	$14,897\;\mathrm{kJ}\cdot {\mathrm{kmol}}^{-3}$
The energy of activation of the propagation reaction $E_{p}$	$3557\;\mathrm{kJ}\cdot {\mathrm{kmol}}^{-3}$
The energy of activation of the termination reaction $E_{t}$	$8.44\times 10^{2}\;\mathrm{kJ}\cdot {\mathrm{kmol}}^{-3}$

**Table 3 polymers-14-00973-t003:** Information of the states in Case Study 2.

Process States	Initial Value	Standard Deviation of Noise	Final Value at Steady State
Dimensionless concentration of initiator ${\tilde{C}}_{I,k}\;$	0.91	0.01	1
Dimensionless concentration of monomer ${\tilde{C}}_{M,k}\;$	0.91	0.01	1
Dimensionless temperature of reactor ${\tilde{T}}_{k}$	0.91	0.01	1

**Table 4 polymers-14-00973-t004:** Information of the parameters in Case Study 2.

Process Parameters	Value
The volumetric flow rates of inlet solvent $F_{s}$	$6.3750\times 10^{-5}\;\mathrm{kmol}\cdot {\mathrm{mol}}^{-3}$
The volumetric flow rates of inlet monomer $F_{m}$	$5.250\times 10^{-5}\;\mathrm{kmol}\cdot {\mathrm{mol}}^{-3}$
The volumetric flow rates of inlet initiator $F_{i}$	$1.55\times 10^{-5}\;\mathrm{kmol}\cdot {\mathrm{mol}}^{-3}$
The volumetric flow rates outlet streams $F_{t}$	$F_{t}=F_{s}+F_{m}+F_{i}$
The reactor volume V	$3{\;m}^{3}$
The residence time τ	$\tau =\frac{V}{F_{t}}$
The measured density ρ	Obtained from simultaneous online measurements
The volumetric heat capacity of reacting mixture $\rho c_{p}$	$1506.24\;\mathrm{kJ}\cdot K^{-1}\cdot m^{-3}$
The heat of reaction due to the propagation reaction $-\Delta H_{p}$	$69,872\;\mathrm{kJ}\cdot {\mathrm{kmol}}^{-3}$
The amount of reaction mixture γ	$\frac{-\Delta H_{p}}{\rho c_{p}}$
The heating rate at the steady state $Q_{ss}$	$-0.6659\;\mathrm{kJ}\cdot s^{-2}$
The inlet feed temperature $T_{i}$	330 K

**Table 5 polymers-14-00973-t005:** Statistical information of the estimated results in Case Study 2.1.

Results	MSE	
	UKF-SPE	UKF-RSPE
States	1.51 × 10^−6^	8.67 × 10^−7^
Parameters	7.09 × 10^−3^	2.08 × 10^−4^
Measurements (MSE = 2.49 × 10^−1^)	1.03	8.58 × 10^−2^

**Table 6 polymers-14-00973-t006:** Statistical information of the estimated results in Case Study 2.2.

Results	MSE	
	UKF-SPE	UKF-RSPE
States	7.17 × 10^−6^	9.20 × 10^−7^
Parameters	4.53 × 10^−2^	6.04 × 10^−3^
Measurements (MSE = 2.59 × 10^−1^)	1.22	2.35 × 10^−1^

## Data Availability

Not applicable.

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
