# Peer review of "Unscented Kalman Filter-Based Robust State and Parameter Estimation for Free Radical Polymerization of Styrene with Variable Parameters"

_polymers, 2022, doi:10.3390/polym14050973_

Round 1

Reviewer 1 Report

This article describes application of Unscented Kalman Filter 

for stylene radical polymerization.

Basically, it is worth accepted as it is, but mention to add  

chemical reactions and condition of products in reactors

and so on, if possible. Anything besides results may 

change depending on reaction condition that were

investigated in this study mathematically.

Additonal comments:   --- The topics (at least viewpoint) of this review seemed to be original one even if  there are other reviews about  polymer synthesis.   The construction of this review article was reasonable to understand for readers,  though it was not short articles. Figures helped to grasp the contents, too.   As it is review, conclusions consistent with the evidence and arguments presented are based on collected papers essentially. However, natural summary was described.

That's all.

Reviewer 2 Report

UKF-based robust SPE method seems to be a promising solution to the problem arising during radical polymerization of styrene. I would like to know if the same system can be used in the case of other radical polymerization processes, or is it dedicated exclusively to the described polymer?

This additional description of possible universal applications should be included in the publication. For this reason, I suggest minor revision.

----------------------------------------------------------------------

1. How original is the topic?
The Kalman filter algorithm is not new, however, its application is still being researched and constitutes a subject of publications, e.g.:
Energies 2021, 14(3), 750; https://doi.org/10.3390/en14030750
Appl. Sci. 2020, 10(3), 850; https://doi.org/10.3390/app10030850,
Journal of Sensors 2021 |Article ID 9002643 | https://doi.org/10.1155/2021/9002643,
Journal of the Taiwan Institute of Chemical Engineers Volume 106, January 2020, Pages 20-33, https://doi.org/10.1016/j.jtice.2019.10.015,
The application of an unscented Kalman filter with an aim to estimate Free Radical Polymerization of Styrene is a new subject. I have not found any prior analysis of this subject.

2. Is the paper well written and easy to read?
In my opinion, the paper is not easy to read. Additional, simple examples of the application of the filter would be welcome.

3. Are the conclusions consistent with the evidence and arguments presented?
Yes, in my opinion, the conclusions were drawn based on the obtained results and are consistent with the presented evidence. 
